# Dysbiosis of the Gut Microbiota and Kynurenine (Kyn) Pathway Activity as Potential Biomarkers in Patients with Major Depressive Disorder

**DOI:** 10.3390/nu15071752

**Published:** 2023-04-03

**Authors:** Ping Lin, Dan Li, Yun Shi, Qingtian Li, Xiaokui Guo, Ke Dong, Qing Chen, Xiaoyan Lou, Zhenhua Li, Ping Li, Weifeng Jin, Shuzi Chen, Yang Sun, Jing Sun, Xunjia Cheng

**Affiliations:** 1Department of Medical Microbiology and Parasitology, School of Basic Medical Sciences, Fudan University, Shanghai 200032, China; linpingsun2000@aliyun.com; 2Department of Clinical Laboratory, Shanghai Mental Health Center, Shanghai Jiao Tong University School of Medicine, Shanghai 200030, China; sichu.youdang@163.com (D.L.); 16111510050@fudan.edu.cn (Y.S.); 16111510044@fudan.edu.cn (Q.C.); louxiaoyan0717@163.com (X.L.); lzhua18@163.com (Z.L.); liping0691@163.com (P.L.); 13512185146@163.com (W.J.); csz0664@163.com (S.C.); 3Department of Microbiology and Immunology, The College of Basic Medical Science, Shanghai Jiao Tong University, Shanghai 200025, China; qingtianli@sjtu.edu.cn (Q.L.); xkguo@shsmu.edu.cn (X.G.); kiwidong@shsmu.edu.cn (K.D.); 4Institute of Arthritis Research in Integrative Medicine, Shanghai Academy of Traditional Chinese Medicine, Shanghai 200052, China; shmzx_sy@aliyun.com; 5Institute for Integrated Intelligence and Systems, School of Medicine and Dentistry, Griffith University, Gold Coast Campus, Gold Coast, QLD 4215, Australia; j.sun@griffith.edu.au; 6School of Computer Sciences, University of Technology Sydney, Sydney, NSW 2007, Australia

**Keywords:** major depressive disorder, gut microbiota, kynurenine, biomarkers

## Abstract

With increasing attention paid to the concept of the microbiota–gut–brain axis, mounting evidence reveals that the gut microbiota is involved in a variety of neurological and psychiatric diseases. However, gut microbiota changes in major depressive disorder (MDD) patients and their association with disease mechanisms remain undefined. Fifty MDD patients and sixty healthy controls were recruited from the Shanghai Healthy Mental Center, China. Fecal samples were collected, and the compositional characteristics of the intestinal flora were determined in MDD patients by MiSeq sequencing. Venous blood was collected for the detection of plasma indoleamine-2,3-dioxygenase (Ido), kynurenine (Kyn) and tryptophan (Trp) levels. Stool samples of bacterial 16S sequencing was carried out. A total of 2,705,809 optimized sequences were obtained, with an average of 54,116 per sample. More unique OTUs were observed at the family, genus and species levels in the control group compared with the MDD cases. Further analysis showed significant changes in the α- and β-diversities and relative abundance levels of gut microbial entities in MDD patients, as well as elevated amounts of Ido and Kyn indicating Kyn pathway activation, KEGG bacterial 16S function prediction analysis shows a variety of amino acids and metabolic (including Ido, Trp and Kyn) changes in the body of patients with MDD. These may result in increased neurotoxic metabolites and reduced generation of serotonin in the disease process. These changed factors may potentially be utilized as biomarkers for MDD in the future, playing more important roles in the disease course.

## 1. Introduction

Major depressive disorder (MDD) is a highly debilitating psychiatric condition, with an estimated yearly prevalence 6% [1] and a female predilection [2,3]. To date, the diagnosis of MDD largely relies on clinical assessments and questionnaires, and more objective biomarkers for predicting and monitoring the disease process are lacking. The disease mechanisms of MDD remain unclear, with genetic and environmental factors shown to be associated with its initiation [4,5]. Various mechanisms such as the monoamine theory, inflammation, changes in the hypothalamic–pituitary–adrenal (HPA) axis, neuroplasticity, neurogenesis, and brain structure and function have been hypothesized to be correlated with MDD’s disease mechanisms [6]. Among these proposed mechanisms, the monoamine theory is one of the most widely studied and has been the main target for treatment, with selective 5-hydroxytryptamine (5-HT) reuptake inhibitors (SSRIs) being the mainstay therapy for MDD [6].

The gut microbiota is involved in various important homeostatic events pertaining to gastrointestinal functions and other complex modulatory processes, including glucose, lipid, microelement and bone metabolism, inflammation and immune response, and peripheral (enteric) and central neurotransmission [7]. Importantly, increasing evidence suggests that the gut microbiota is involved in neurological and psychiatric diseases [8]. The concept of the microbiota–gut–brain axis started to attract increased attention in the early 2000s, with researchers proposing that the gut microbiota and its metabolites may affect brain functions and behavior through the microbiota–gut–brain axis in a bidirectional interplay via direct and indirect pathways, which involve the neuroendocrine system, the HPA axis, and various other pathways [8]. In addition, animal models have shown that fecal transplantation from individuals with depression results in increased depressive-like behaviors [9,10].

The microbial flora plays a causal role in physical diseases, suggesting that abundance variation in the gut microbiota may cause depression and anxiety through the microbiota–gut–brain axis [11]. Human studies have shown that administration of probiotics affects mood and alleviates anxiety in healthy cohorts [12,13]. Importantly, a few human studies have shown differences in microbiota composition in MDD cases compared with controls [10,14]. However, such differences were not detected by other studies [15]. Therefore, gut microbiota changes in MDD patients remain need to be further determined.

The gut microbiota produces various neurotransmitters, microbial metabolites, and amino acids that interact directly or indirectly with the brain. In addition, evidence suggests that the gut microbiota potentially correlates with tryptophan (Trp) metabolism. Trp is an essential aromatic amino acid that plays a critical role in protein synthesis. It can be converted to serotonin, although most Trp is degraded via the kynurenine (Kyn) pathway [16], leading to the generation of catabolites, including Kyn and its subsequent metabolites such as quinolinic acid and kynurenic acid. Importantly, Kyn and its metabolites have been linked to MDD [17,18], and the Trp to Kyn pathway is regulated by indoleamine-2,3-dioxygenase (Ido) as the rate-limiting enzyme [19,20] that can be activated under various conditions (Figure 1).

Germ-free (GF) and antibiotic-treated mice show increased levels of circulating total Trp [21,22], and the gut microbiota is capable of stimulating Ido activity in animal models [23]. In addition, in vitro and in vivo studies demonstrated that *Lactobacillus johnsonii* reduces Ido activity in intestinal epithelial cells while decreasing serum Kyn levels [24], further implicating the gut microbiota in Trp metabolism. Therefore, the present study aimed to characterize gut microbiota changes and determine the potential role of the Kyn pathway in MDD. Understanding gut microbiota changes and Trp and Kyn pathway regulation in MDD may provide novel insights into potential disease mechanisms and help determine potential biomarkers of MDD.

## 2. Materials and Methods

### 2.1. Cohorts

This study was approved by the Institutional Review Board (IRB) of Shanghai Healthy Mental Center (No. 2020ky-13), and all participants provided written informed consent in line with the Declaration of Helsinki prior to enrolment. The inclusion criteria were: (1) 20–85 years of age; (2) no use of antibiotics, antifungals, or probiotics or probiotic-related drinks within the last month. In the MDD group, diagnosis (performed independently by two specialists) was based on the major depressive disorder criteria in the Diagnostic and Statistical Manual of Mental Disorders (DSM-IVTR), with 17-item Hamilton Depression Scale (HAM-D) scale scores ≥23 required for inclusion. The control group included individuals in good mental and physical health with no stomach/gut problems such as chronic diarrhea, constipation, gas, heartburn, or bloating. Additionally, the control patients must have not been related to any individual with MDD. Participants with current infections or autoimmune disorders (current or previously diagnosed) that could influence the inflammatory state were excluded. Fifty MDD patients and 60 healthy controls were recruited from the Shanghai Healthy Mental Center, China. The demographic features are detailed in Table 1.

### 2.2. Fecal Sample Collection and Handling

Fecal samples (about 1 g/participant) were collected from the whole cohort on the day of enrolment and transported in liquid nitrogen. The samples were subsequently stored at −80 °C until DNA extraction.

### 2.3. DNA Extraction and Amplification of V3–V4

Total bacterial DNA was extracted from 0.5 g of each fecal sample using the QIAamp DNA Stool Mini Kit (Qiagen, Duesseldorf, Germany) by following the manufacturer’s instructions. DNA quantity and quality were assessed by measuring the absorbance at 260 and 280 nm with a NanoDrop ND-2000 spectrophotometer (NanoDrop Technology, Rockland, DE, USA). 16S rDNA high-throughput sequencing was performed using a MiSeq instrument (Illumina Inc., San Diego, CA, USA). The V3–V4 regions of the 16S rDNA gene were amplified using barcode-indexed primers (Shanghai Sangon, China), including 338F and 806R, with 5′ACTCCTACGGGAGGCAGCA-3′ and 5′-GGACTACHVGGGTWTCTAAT-3′ as the forward and reverse sequences, respectively.

### 2.4. MiSeq Sequencing

Amplicons were purified from a 2% gel using an AxyPrep DNA Gel Extraction Kit (Axygen Biosciences, Union City, CA, USA) as per the manufacturer’s instructions. The concentrations of polymerase chain reaction (PCR) products were measured with QuantiFluorTM-ST (Promega, Madison, WI, USA), then normalized at equimolar concentrations and subjected to paired-end sequencing (2 × 250) on the PE300 platform of the MiSeq instrument (Illumina Inc., San Diego, CA, USA), following standard protocols.

### 2.5. Operational Taxonomic Units (OTUs) Clustering and Annotation

Raw paired-end reads with overlapping nucleotides were assembled using the PANDAseq program after sequencing, and subsequently quality-filtered as follows. Bases with tail quality values below 20 were filtered out with a 50 bp window. In cases where the average quality value in the window was less than 20, back bases were cut from the window. Reads less than 50 bp were filtered after quality control, with those containing N bases being removed. Paired reads with a minimum overlap length of 10 bp were merged into one sequence. The detection parameters had been standardized according to the manufacturer’s requirements, the maximum allowable mismatch ratio in the overlap region of a spliced sequence was 0.2. Unmatched sequences had been screened. All samples were distinguished according to barcodes and primers at the beginning and end of the sequence by adjusting the sequence direction. No mismatching was allowed in the barcode, and the maximum number of primer mismatches was two. Fastp and FLASH software were used for analysis [25].

OTUs were determined using a 97% similarity cut-off, followed by clustering analysis using Usearch (version 7.0, http://drive5.com/uparse/ (accessed on 7 December 2022)). Specific taxonomic information corresponding to each OTU was obtained from the Silva database (Release132, http://www.arb-silva.de (accessed on 7 December 2022)).

### 2.6. Fecal Microbiome Analysis

In this study, α-diversity was measured as the richness of species based on the rarefied OTU table, and β-diversity was estimated by computational analysis of unweighted UniFrac scores; a principal coordinate analysis (PCoA) was applied to the distance matrices for visualization and plotted against each other to summarize the microbial community compositional differences between the samples [26]. Hierarchical cluster analysis was carried out using the single-linkage, complete-linkage, and average-linkage clustering methods.

### 2.7. Quantitative Analysis of 16S rRNA Genes

Quantitative PCR (qPCR) was validated by constructing an artificial mixture of 16S rRNA genes using published approaches [27,28]. Bacteroide-specific primers (Shanghai Sangon, Shanghai, China) were utilized: forward primer, 5′-CATGTGGTTTAATTCGATGAT-3′; reverse primer, 5′-AGCTGACGACAACCATGCAG-3′.

### 2.8. Blood Collection and Handling

Participants fasted for at least 8 h before blood collection into EDTA tubes between 0730 AM and 0900 AM in the supine position. EDTA tubes containing blood were turned upside down 10–15 times and centrifuged immediately at 2000 rpm for 15 min. The plasma was collected, aliquoted into five Nunc glass tubes, and stored at −80 °C until use.

### 2.9. Detection of Plasma Ido, Kyn, and Trp

Ido, Kyn, and Trp levels were measured by a sandwich enzyme-linked immunosorbent assay using commercially available kits (Jianglai Industrial, Shanghai, China) as per the manufacturer’s instructions.

### 2.10. Statistical Analyses

Statistical analyses were performed with R (v.3.2.2). The Wilcoxon rank-sum test was performed to assess differences in α-diversity between the two groups. The Mann–Whitney *U* test was performed to compare continuous variables between groups, and a multiple test correction was performed using the Benjamini and Hochberg false discovery rate. In addition, Spearman’s rank sum test was performed for the correlation analysis.

## 3. Results

### 3.1. Demographic Information of the Participants and Obtained Sequences

A total of 110 samples from 50 MDD patients and 60 healthy controls were analyzed in this study. The demographic features of all subjects, including their disease durations, are detailed in Table 1. A total of 2,705,809 optimized sequences were obtained, with an average of 54,116 per sample and an average sequence length of 415 bp.

### 3.2. α-Diversity of the Gut Microbiome

The Sobs, Chao, and Shannon indexes revealed that the α-diversity differed significantly at the family, genus, and species levels between the MDD and healthy control groups (Figure 2), with a higher gut flora diversity in the control group than in the MDD group, indicating an association between a decreased α-diversity of the intestinal microbiome and MDD development.

### 3.3. β-Diversity of the Gut Microbiome

PCoA of the OTU abundance levels revealed that the β-diversity was significantly higher in the control subjects than in the MDD patients. In addition, the first and second coordinate percentages of bacterial community variation based on the PCoA were 14.17 and 13.03%, respectively (Figure 3A). Furthermore, the partial least squares discriminant analysis (PLS-DA) of the OTU levels showed that the MDD and control groups were separated in the direction of the COMP1 axis, with a value of 34.77% (Figure 3B).

### 3.4. Alterations in the Abundance of the Gut Microbiome

Based on the Venn diagram, no unique phylum was identified in the MDD group; however, 13 unique phyla were detected in the control group and another 13 phyla were present in both groups. More unique OTUs were also observed at the family, genus, and species levels in the control subjects than in MDD patients (Figure 4).

In general, bacteria in the phyla Firmicutes, Bacteroidetes, Proteobacteria, Actinobacteria, and Verrucomicrobia are the most abundant in the human intestine. Community bar plot analysis revealed increased proportions of Proteobacteria, Actinobacteria, and Verrucomicrobia and decreased proportions of Firmicutes and Bacteroidetes in MDD patients compared to in controls (Appendix A). The Wilcoxon rank-sum test plot revealed that 13 species differed significantly in abundance at the phylum level between MDD and control cases. In the MDD group, the most significantly upregulated phylum was Actinobacteria (*p* < 0.001), followed by Verrucomicrobia and unclassified_k_norank_d_bacteria (*p* < 0.01), as well as Synergistetes and Proteobacteria (*p* < 0.05). Compared with the control group, the significantly down-regulated phyla in the MDD group included Bacteroidetes, Patescibacteria, Tenericutes, Epslionbacteraeota, Cyanobacteria, Acidobacteria, Deferribacteres, and Spirochaetes (*p* < 0.001; Appendix A).

At the family level, nine species from the top 15 families had significant differences in abundance between the MDD and control cases. The up-regulated species in MDD patients belonged to the Enterobacteriaceae, Bifidobacteriaceae, Enterococcaceae, and Streptococcaceae families, while the down-regulated species belonged to the Ruminococcacceae, Prevotellaceae, Muribaculaceae, Lactobacillaceae, and Rikenellaceae families (Appendix A). At the genus level, 11 species from the top 15 genera presented significant differences in abundance between the MDD and control cases. Specifically, the increased genera included *Blautia*, *Escherichia-Shigella*, *Bifidobacterium*, and *Ruminococcus torques*, while the decreased genera included *Faecalibacterium*, *Prevotella*, *Agathobacter*, *Lactobacillus*, *norank_f_Muribaculaceae*, *Lachnoclostridium,* and *Megamonas* (Appendix A).

In this study, we found many changes in the subjects’ group using 16S flora analysis for predicting function. The experimental group had a remarkable difference with that in control group involved in cell cycle, division, chromosome partition, nucleotide, amino acid, carbohydrate, inorganic ion transport, metabolism and energy production and conversion with KEGG orthologous (Wilcoxon rank-sum test FDR < 0.05, Figure 5A, Appendix A). In the group of patients with MDD this was significantly higher than the control group for amino acid, carbohydrate, inorganic ion transport, metabolism, energy production and conversion. Indole amine is a kind of biogenic amine belonging to a single amine neurotransmitter, participating in supplying blood for the brain in the central nervous system. One major indole amine substance includes tryptophan (Trp) indoleamine-2,3-dioxygenase (Ido). The tryptophan–kynurenine pathway (KP) items (Kyn, Trp, Ido) are important amino acids and metabolic products in the body directly related to the MDD. As a result, their significant change may be potential biomarkers of the disease in prediction and diagnosis MDD in the future (Figure 5B). Subject to the sample size, the results are not perfect, the team will explore the unknown function further in future work.

### 3.5. Plasma Ido, Kyn, and Trp Levels

Plasma Ido levels were significantly elevated in MDD cases compared to the controls (*p* < 0.001; Figure 6A), indicating increased Ido activation and subsequent activation of the Kyn pathway. As expected and shown in Figure 6B, plasma Kyn levels were also higher in MDD cases than in the controls (*p* < 0.001), consistent with previous findings [29]. However, there was no significant difference in the plasma Trp levels between the MDD and control groups (Figure 6C).

## 4. Discussion

This study aimed to assess gut microbiota changes and Kyn pathway activation in human MDD cases. We found a different abundance in their gut microbiota between healthy controls and MDD patients, and these subjects also showed Kyn pathway activation, which may be involved in the pathological mechanisms of MDD.

In humans, 95% of microorganisms are located in the intestine, and they act by regulating various functions and participating in the gut and systemic immune responses [30]. In addition, gut microbiota dysbiosis has been shown to be closely correlated with psychological behavior and diseases in recent years [8]. A clinical study found that depression is often accompanied by damage to mucosal epithelial cells, changes in intestinal mucosa permeability, and the translocation of intestinal bacteria [7], all of which are suggestive of potential microbiota changes in MDD. Although animal studies have demonstrated a significant contribution of the gut microbiota to MDD, only a few human studies have been carried out to evaluate fecal microbiota changes in MDD, with inconsistent results across the studies. Despite the different individual results, the abundance levels of several major phyla significantly differ between MDD and healthy controls [31]. As shown above, at the genus level, *Blautia*, *Escherichia Shigella*, and Bifidobacterium were more abundant in MDD patients who had reduced *Faecalibacterium* and *Prevotella* levels compared to the controls. Authors of a meta-analysis [32] have suggested that the different conclusions of previous studies are controversial. For example, the increased abundance of *Escherichia coli* in patients with depression may indicate that there have differences in the composition of different populations within *Escherichia*. In addition, when the sample size is small, the possibility that some genera of *Proteobacteria*, including *Eschella*, may constitute inconsistent abundances. The study of intestinal flora is complex and may be influenced by many factors, including the environment, race, age, sex, and diet, as well as the number of patients studied, the inclusion criteria, and whether the retention of samples is standardized. So it is quite normal that different studies come to different conclusions, and eliminating these differences is one of the future goals of many teams including us.

The present study also revealed that at the phylum level, there was a significantly increased relative abundance of *Proteobacteria*, *Actinobacteria*, and *Verrucomicrobia* and a significantly decreased relative abundance of *Firmicutes* and *Bacteroidetes* in MDD cases compared to controls, providing further evidence of the role of gut microbiota dysbiosis in MDD.

Changes in the diversity of the intestinal microflora in patients with depression are controversial. Some authors reported a decrease in α-diversity [33], while others reported an increase [14] or even no difference [15]. The present study assessed the α- and β-diversity of the fecal microbiota in MDD cases and revealed significant changes in 13 species compared to the controls, supporting that gut microorganisms may serve as potential biomarkers of MDD. Such changes may also suggest the potential role of the gut microbiota in the pathological mechanisms of MDD.

Human studies have shown that administration of *Lactobacillus plantarum* 299v, in addition to SSRI treatment in MDD patients, significantly reduces circulating Kyn concentrations compared with SSRI treatment alone [34], indicating the involvement of the microbiota in the disease process. In addition, lipopolysaccharides (LPS) may decrease the frequency of positive moods and cause depression-like behaviors [35,36]. The present study showed an increased abundance of Gram-negative bacteria that produce LPS in MDD cases, whereas the abundances of protective organisms such as *Lactobacillus* and *Faecalis* were decreased. Taken together, these results suggest that microbiota dysbiosis may participate in the pathogenesis of MDD. Interestingly, in the control group, the bacterial abundance in some fecal samples was similar to those of the patients with MDD in this study. In contrast, the control individuals had completely distinct microbial compositions compared to the MDD group, suggesting a floral diversity within controls; however, larger cohort studies are required to confirm these results.

It is widely admitted that serotonin deficiency or serotonin receptor dysfunction is involved in the pathogenesis of MDD [37]. Although serotonin synthesis mainly occurs in the gut [38], Trp could cross the blood–brain barrier and lead to serotonin synthesis in the brain [39]. Ido, the rate-limiting enzyme of the Kyn pathway, is distributed in various extrahepatic tissues, including the brain [40] and particularly in glial cells [39]; therefore, in the central nervous system (CNS), Kyn metabolism mainly occurs in glial cells. Within the CNS, inflammatory factors have been shown to activate Ido in glial cells [30,31], thereby inducing the Kyn pathway to generate neurotoxic metabolites [41,42].

Quinolinic acid accumulation in the brain can over-activate the receptors on glial cells, producing excitatory toxicity, enhancing oxidative stress, and promoting the initiation and development of diseases [43]. Furthermore, pro-inflammatory cytokines may increase the expression of Ido [44] and induce its activation [45], leading to enhanced Kyn pathway activation, competitively reducing Trp available for serotonin generation [46]. The present study revealed Ido and Kyn elevations in MDD patients were indicative of Kyn pathway activation. Although other researchers have revealed decreased levels of Trp in MDD cases [47], others have shown increased plasma concentrations of Trp [48]. This study detected no significant changes in plasma Trp levels in MDD cases, showing that Ido and Kyn pathway activation may be a more sensitive and stable indicator of MDD than Trp changes.

The roles of the gut microbiota in regulating Trp catabolism and the Kyn pathway remain unclear. Both human and animal studies have demonstrated that systemic administration of bacterial LPS increases cytokine levels, decreases positive moods, and induce depression-like symptoms [35,36,49]; in agreement, animal studies have revealed that administration of LPS induces Ido expression in different brain regions, resulting in depressive-like behaviors [50,51]. Therefore, the gut microbiota may participate in MDD pathogenesis by inducing inflammation and subsequent activation of Ido, resulting in increased neurotoxic metabolites and reduced serotonin production (Figure 6D).

The limitations of this study should be mentioned. First, this was a single-center study with a limited sample size. Therefore, the data have low generalizability. In addition, the causal relationship between the gut microbiota changes observed and Kyn pathway activation was not demonstrated. Further studies are warranted to resolve these shortcomings.

## 5. Conclusions

In conclusion, this study showed changes in the diversity and relative abundance of the gut microbiota in MDD patients, as well as elevated Ido and Kyn concentrations indicative of Kyn pathway activation, which may be utilized as biomarkers of MDD. Furthermore, gut microbiota dysbiosis may not only reflect a change in MDD but also play a role in the pathological mechanism of this disease.

## Figures and Tables

**Figure 1 nutrients-15-01752-f001:**
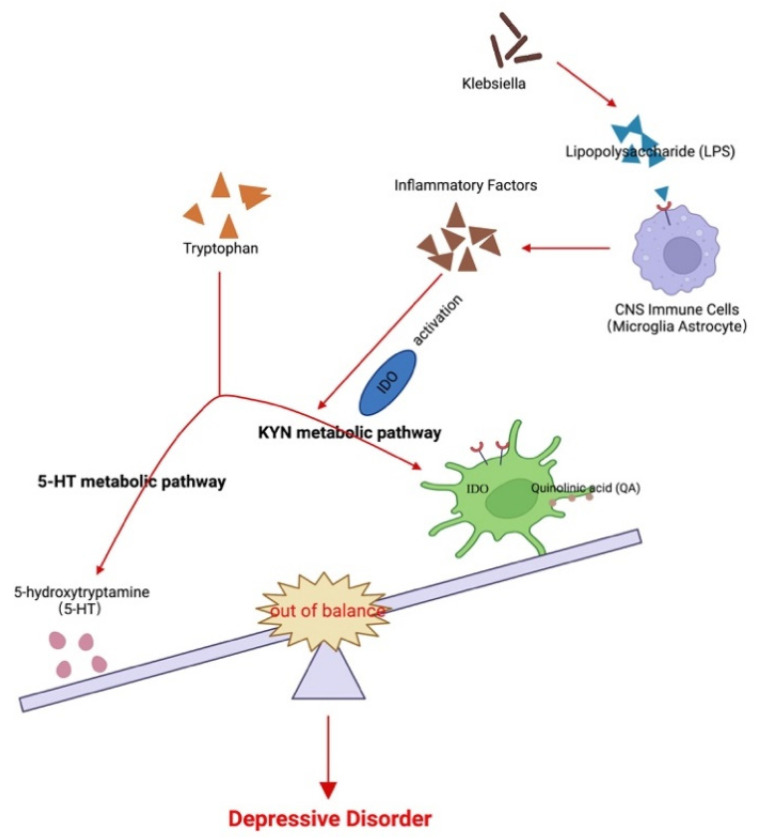
Possible association among the gut microbiota, kynurenine pathway, and major depressive disorder. Changes in the gut flora lead to LPS and other factors to induce CNS immune cells to secrete inflammatory factors and activate Ido. This, in turn, leads to an imbalance of tryptophan metabolites between 5-HT and the Kyn metabolic pathways. The imbalance of the products of the two pathways, 5-HT and quinolinic acid, further participate in the pathogenesis of MDD.

**Figure 2 nutrients-15-01752-f002:**
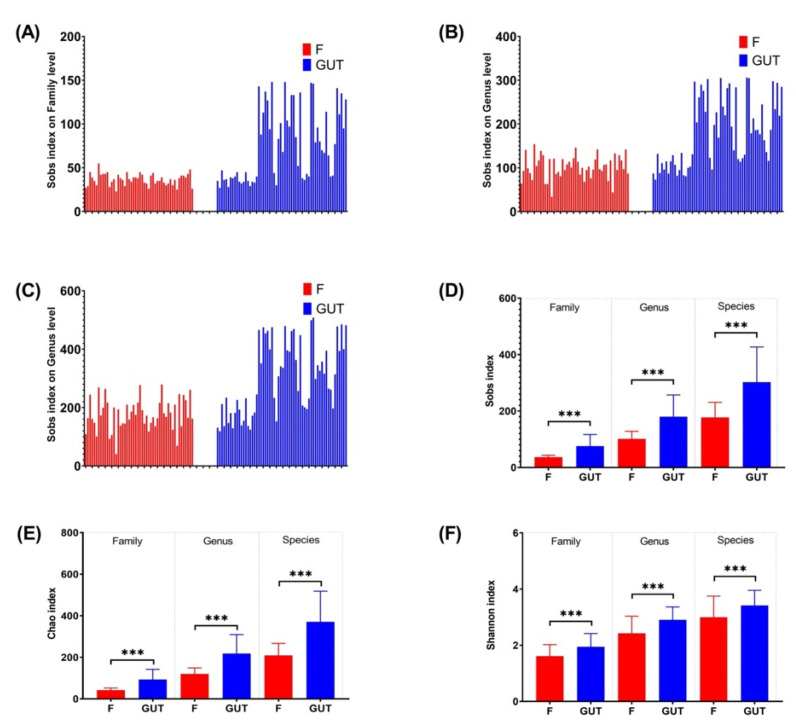
α-diversities of the gut microbiome in MDD and control cases. α-diversities were estimated at the family (**A**), genus (**B**) and species (**C**) levels by the Sobs index. The Wilcoxon rank-sum test was performed to compare Sobs index (**D**), Chao index (**E**) and Shannon index (**F**) values of OTUs at the family, genus and species levels between the MDD (F, red bars) and control (GUT, blue bars) groups. Data are presented as the mean ± standard deviation. *** *p* < 0.001.

**Figure 3 nutrients-15-01752-f003:**
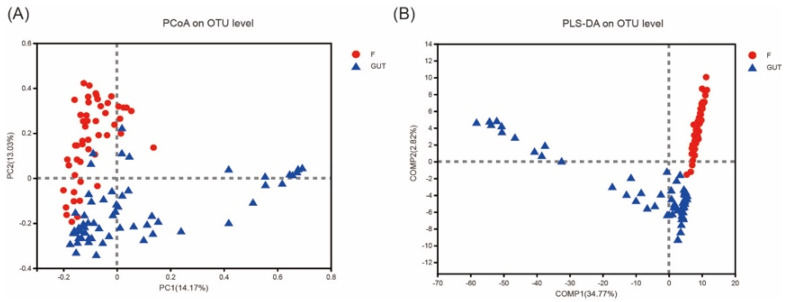
β−diversities of the gut microbiome in MDD and control cases. (**A**) Principal component analysis (PCoA) plot generated from the weighted UniFrac analysis. The x−axis indicates the first coordinate (PC1), and the y−axis indicates the second coordinate (PC2). The values in parentheses are the percentages of the community variation explained. The red and blue dots depict microbial enrichment in the MDD (F) and control (GUT) groups, respectively. (**B**) Partial least squares discriminant analysis (PLS−DA) of the MDD (F, red) and control (GUT, blue) groups.

**Figure 4 nutrients-15-01752-f004:**
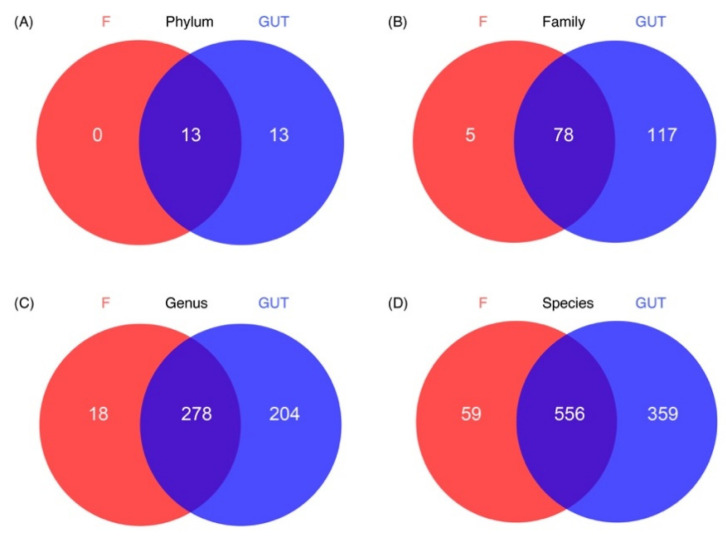
Characteristics of the gut microbiome in the MDD and control cases. Venn diagrams show the unique OTUs in the MDD (F, red) and control (GUT, blue) groups at the phylum (**A**), family (**B**), genus (**C**) and species (**D**) levels, as well as the number of overlapped OTUs between the two groups.

**Figure 5 nutrients-15-01752-f005:**
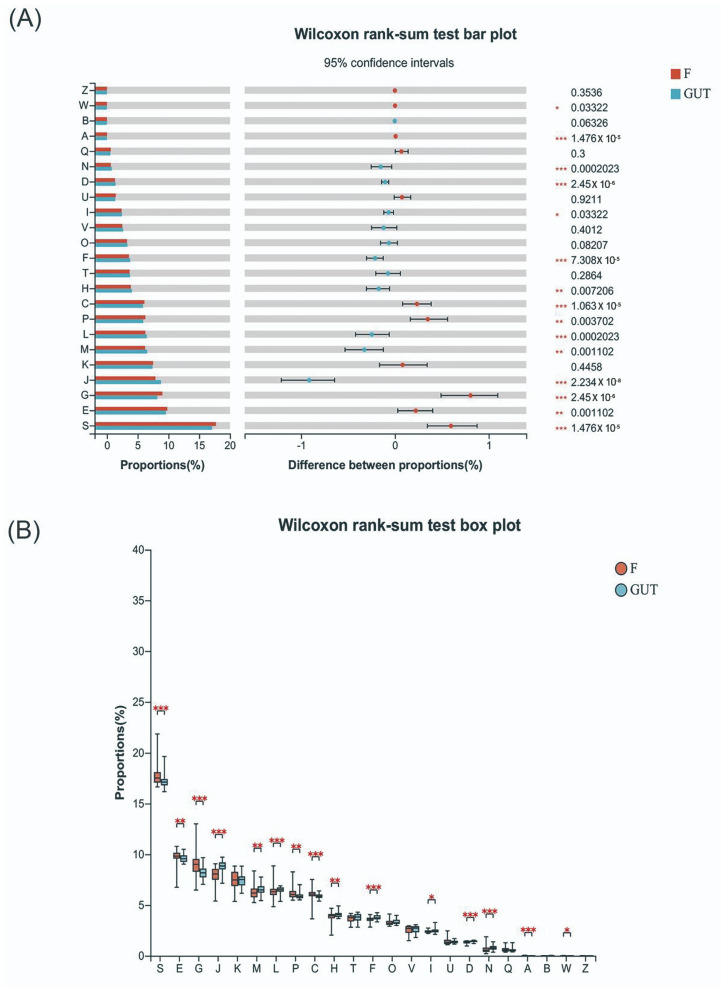
Tryptophan–kynurenine pathway items may be potential biomarkers of MDD. (**A**) KEGG orthologous analysis found significant differences in cell cycle control, cell division and chromosome partition, the transportation and metabolism of nucleotides, amino acids, carbohydrates, and inorganic ions, and energy production and conversion between MDD patients and the control group. (**B**) Indole amines, such as Trp, Ido, and Kyn, are key metabolic products related to MDD. S—function unknown; E—amino acid transport and metabolism; G—carbohydrate transport and metabolism; J—translation, ribosomal structure and biogenesis; M—cell wall/membrane/envelope biogenesis; L—replication, recombination and repair; P—inorganic ion transportation and metabolism; C—energy production and conversion; H—coenzyme transportation and metabolism; F—nucleotide transportation and metabolism; D—cell cycle control, cell division, and chromosome partitioning; N—cell motility; A—RNA processing and modification.

**Figure 6 nutrients-15-01752-f006:**
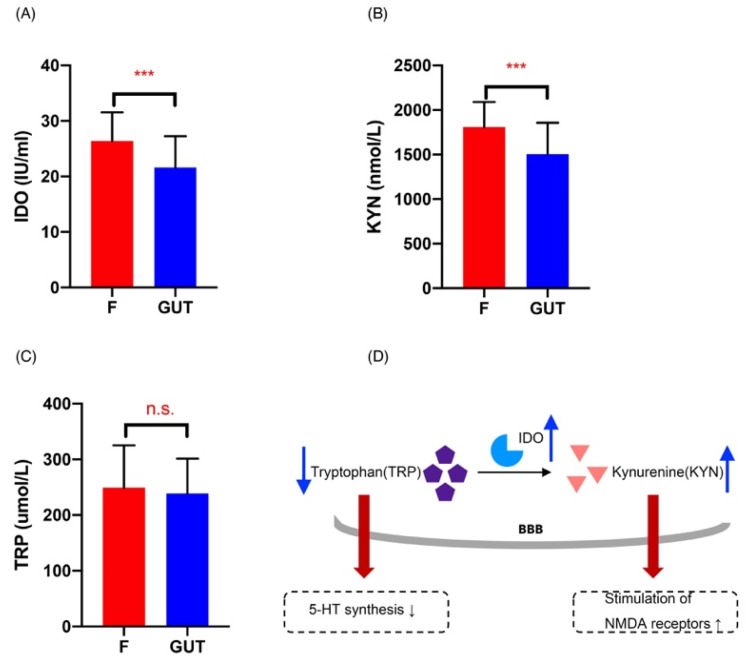
Changes in the Kyn pathway in MDD patients. Plasma Ido (**A**), Kyn (**B**), and Trp (**C**) concentrations in the MDD (F, red) and control (GUT, blue) groups, revealing significantly increased Ido (**A**) and Kyn (**B**) in MDD cases compared with the controls. (**D**) Summary of Kyn pathway changes in MDD patients: increased Ido activity induces Trp metabolism, resulting in: (i) increased production of Kyn that causes hippocampal damage and oxidative stress by overstimulating NMDA receptors, and (ii) the decreased availability of Trp that crosses the blood–brain barrier (BBB), leading to a reduced synthesis of 5-HT. Data are presented as the mean ± standard deviation. *** *p* < 0.01 versus the control group.

**Table 1 nutrients-15-01752-t001:** Age and sex of the assessed study subjects.

Group	Gender	Age	Number of Subjects
MDD (N = 50)	Female (N = 36)	10–19	6
		20–29	8
		30–39	3
		40–49	4
		50–59	5
		60–69	5
		70–79	3
		≥80	2
	Male (N=14)	10–19	4
		20–29	3
		30–39	4
		60–69	2
		≥80	1
control (N = 60)	Female (N = 31)	20–29	5
		30–39	2
		40–49	11
		50–59	10
		60–69	3
	Male (N = 29)	20–29	9
		30–39	4
		40–49	6
		50–59	6
		≥60	4

## Data Availability

The nucleotide sequences of each strain genome in this study are available in the Chinese CNCB database (https://ngdc.cncb.ac.cn/gsub/ (accessed on 7 December 2022)) under accession number: PRJCA009406.

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
