# Peer review of "Dysbiosis of the Gut Microbiota and Kynurenine (Kyn) Pathway Activity as Potential Biomarkers in Patients with Major Depressive Disorder"

_nutrients, 2023, doi:10.3390/nu15071752_

Round 1

Reviewer 1 Report

This study is well presented, and I have only a few minor comments:

- all the information from the Results section that is not connected to the results obtained from this investigation should be transfered to more appropriate sections (Introduction or Discussion)

- figures could be re-uploaded with better resolution, as some of them are really hard to read and hence understand

-  in Discussion section, more emphasis should be put on potential clinical and future implications that can be derived from the results from the study

Author Response

Dear reviewer:

thank you very much for your advise,we have revised manuscript

Reviewer 2 Report

The main finding of the study is the assessment of gut microbiota changes in human patients with major depressive disorders. Moreover, the potential role of the Kyn pathway in major depressive disorders it has also been highlighted.

I think that the subject of the paper is very interesting in the current panorama, due to the increasing evidence about the involvement of the gut microbiota in neurological and psychiatric disease. Introduction, methodology, and discussion are fully appropriate, and the findings of this paper could be the first step to identify new biomarkers in MDD patients.

However, according to my opinion there are some limitations: first, it would be very important if the authors could expand the number of patients involved in the study, verifying the results in a larger sample size. This will also allow the authors to conduct a separate analysis in men and women, improving the quality of work with a gender analysis, that represent a key issue.

Moreover, the manuscript needs a language revision by a native English speaker, particularly in the results Section. Just for example, on page 8 (lines 253-263) there are grammatical mistakes, and some sentences are difficult to understand; on page 9 (lines 265-266) the authors wrote: “KEGG orthologous analysis found significant differences were found….

A native English speaker, improving the clearness of the text would definitely increase the quality of the work.

Minor comments:  

Please, specify the acronyms CNS, IDO on page 3 (Legend of the Fig 1) to facilitate the reader

Please, improve the quality of Fig. 5

Author Response

Dear editor:

Thank you very much for your advise, we have finished revised. Thank you very much again!
